# Change in Shoulder Function in the Early Recovery Phase after Breast Cancer Surgery: A Prospective Observational Study

**DOI:** 10.3390/jcm10153416

**Published:** 2021-07-31

**Authors:** Jihee Min, Jee Ye Kim, Sujin Yeon, Jiin Ryu, Jin Joo Min, Seho Park, Seung Il Kim, Justin Y. Jeon

**Affiliations:** 1Department of Physiology, Yonsei Institute of Sports Science & Exercise Medicine, Yonsei University Wonju College of Medicine, Wonju 26426, Korea; jihee8700@yonsei.ac.kr (J.M.); seuyeonn@naver.com (S.Y.); 2Exercise Medicine and Rehabilitation Laboratory, Department of Sport Industry Studies, Yonsei University, Seoul 03722, Korea; wldls82@yuhs.ac (J.R.); wlswn09155@gmail.com (J.J.M.); 3Division of Breast Surgery, Department of Surgery, Yonsei University College of Medicine, Seoul 03722, Korea; JEEYE0531@yuhs.ac (J.Y.K.); PSH1025@yuhs.ac (S.P.); SKIM@yuhs.ac (S.I.K.); 4Exercise Medicine Center for Diabetes and Cancer Patients, Yonsei University, Seoul 03722, Korea; 5Cancer Prevention Center, Yonsei Cancer Center, Shinchon Severance Hospital, Seoul 03722, Korea

**Keywords:** breast cancer, mastectomy, breast-conserving surgery, range of motion, strength

## Abstract

Breast cancer surgery significantly affects the shoulder’s range of motion (ROM) and strength. However, the extent of shoulder impairment, as well as patterns of recovery immediately after surgery, is not fully understood. Therefore, we aimed to investigate shoulder ROM and strength during the early recovery phase after surgery. Thirty-two breast cancer patients were observed five times: the day before surgery, discharge day (postoperative day 1 (POD1) or (POD2)), first outpatient visit (POD7–10), second outpatient visit (POD14–20), and third outpatient visit (POD21–30). We assessed shoulder passive ROM and strength for both affected and unaffected arms at each observation. ROM decreased in both affected and unaffected sides post-surgery. ROM on the affected side did not recover to the pre-surgery level until the third outpatient visit (POD24). In contrast, the ROM on the unaffected side recovered to the pre-surgery level by the first outpatient visit (POD10). The shoulder strength of both arms declined and did not recover to pre-surgery levels. Shoulder strength in the affected arm significantly decreased immediately after surgery (52.9% of the pre-surgery levels) and did not recover until the third outpatient visit (62.5% of the pre-surgery levels), whereas that in the unaffected arm decreased gradually (83.1 ± 2.3 at POD 1 and 78.9 ± 2.9 at POD 24). Descriptively, patterns of recovery in ROM may vary according to types of surgery while patterns of recovery in shoulder strength did not: shoulder strength significantly decreased and did not recover notably regardless of types of surgery. Both shoulder ROM and strength reduced during the early recovery phase after breast cancer surgery regardless of types of surgery, although the degree of reduction was greater in shoulder strength than ROM. Our findings suggest that rehabilitation exercises should be implemented in both upper limbs.

## 1. Introduction

A surgical approach is the primary step of breast cancer treatment, which aims to completely remove the entire tumor [1]. Even though surgical techniques have improved significantly, breast cancer patients still experience adverse effects, including reduced shoulder range of motion (ROM), impaired upper body strength, chronic pain, and sensory disturbances [2,3,4].

One of the most common complications after breast cancer surgery is a functional limitation of the upper body. Up to 67% of breast cancer patients experience arm or shoulder impairment, including pain, numbness, loss of strength, and reduced ROM, after surgery [2,5]. Although most breast cancer patients experience some degree of discomfort and functional limitation, the degree of morbidities after breast cancer surgery varies according to surgical methods. Patients undergoing a mastectomy experience 5.7 times (odds ratio (OR) 5.7, 95% confidence interval (CI) 1.03–31.2) greater postoperative shoulder problems than patients undergoing breast-conserving surgery [5]. Axillary dissection also significantly contributes to the reduction in arm mobility compared with sentinel lymph node biopsy (SLNB) [6]. Regardless of the surgical method, most breast cancer patients experience chronic arm or shoulder discomfort, which lasts up to 3 years after surgery [5].

Shoulder problems in breast cancer patients affect their daily activities, such as pulling a sweater overhead, fastening a bra, zipping up a back zipper, reaching overhead, and carrying heavy bags [7]. A systematic review suggested that at least 150° elbow flexion and 130° shoulder flexion and abduction are required to perform personal care, eating, and drinking [8]. However, many breast cancer patients are unable to achieve 150° elbow flexion and 130° of shoulder flexion even after several years post-surgery [9,10]. Additionally, shoulder morbidities, including pain, loss of strength, and limited shoulder ROM, lead to a declining quality of life in breast cancer patients after surgery [3,11]. Consequently, only 59% of breast cancer survivors returned to work, and even among those who returned to their workplace, a substantial percentage of them were not able to work full time because of physical conditions, including these shoulder problems [12]. Therefore, rehabilitation after breast cancer surgery is essential to improve patients’ quality of life and increase the return to work rate.

Breast cancer patients experience the most frequent and significant shoulder morbidities immediately after surgery. Although studies have reported the long-term consequences of breast cancer surgery in the upper body, shoulder ROM and strength during the early phase of rehabilitation have not been studied. More importantly, variances in shoulder ROM and strength according to the types of surgery are not yet fully understood.

Therefore, the purpose of the current study is to examine the shoulder ROM and strength in both the affected and unaffected arm after breast cancer surgery for up to 4 weeks.

## 2. Materials and Methods

### 2.1. Participants

Thirty-two breast cancer patients were recruited from Severance Hospital, Yonsei University Health System, Seoul, Korea, from 14 February to 2 November 2019. Eligibility criteria included the following: (1) age between 19 and 70 years; (2) histologically confirmed stage <IV breast cancer; (3) ability to understand and provide written informed consent in Korean. We excluded (1) patients who were scheduled bilateral breast surgery, (2) breast reconstruction surgery, or (3) existing evidence of recurrent or metastatic diseases. This study was approved by the Institutional Review Board of Severance Hospital (IRB No. 4-2018-1094), and all participants provided written informed consent.

### 2.2. Study Designs

This study was a prospective observational study. Eligible participants were evaluated for all assessment variables five times over the span of a month (day before surgery, hospital discharge (postoperative day 1, POD1), first outpatient visit (POD7–10), second outpatient visit (POD14–20), and third outpatient visit (POD21–30).

### 2.3. Outcome Measures

All measurements were performed in duplicate by a single investigator, on both the affected and unaffected upper limbs. If the difference between the first and second measurements was >5%, a third measurement was performed, and the closest two values were averaged.

#### 2.3.1. Range of Motion

In each participant, three shoulder movements (flexion, abduction, and extension) of passive range of motion were measured using a digital goniometer (Goniometer bending iron 29-5900, Pakistan). Flexion and abduction were evaluated in the supine position and extension was measured in standing position [13]. 

#### 2.3.2. Shoulder Strength

The shoulder muscle strength, measured in pounds (lb), was determined using a handheld dynamometer (J-tech Medical Industries Inc., Heber City, UT, USA). Strength peak muscle force was measured using maximal voluntary isometric contraction (MVIC) in flexion, abduction, extension [14].

#### 2.3.3. Shoulder Function Score

Shoulder function score was calculated for ease of interpretation of shoulder ROM and strength. It was converted to 100 scales for each measurement relative to the baseline and each assessment summed ROM (flexion+ extension+ abduction) and strength value (flexion+ extension+ abduction) was converted to a standard 100-point scale.

### 2.4. Statistical Analysis

We analyzed the entire data using a parametric method after testing for the normality of distribution using the Shapiro–Wilk test. Descriptive analyses were used to evaluate demographic information, baseline body composition, and medical information. Changes in shoulder ROM, strength, and shoulder function score were assessed for both the affected and unaffected sides using univariate repeated measures of ANOVA (RM-ANOVA). When the difference was statistically significant, we conducted a post-hoc comparison between the baseline evaluation and each measurement point. The post-hoc test was performed using the paired *t*-test with Bonferroni’s correction [15]. After baseline value adjustment, analysis of covariance was used to compare differences between the affected and unaffected arms. RM-ANOVA was conducted to investigate the pattern of changes according to the surgical type in shoulder ROM, strength, and shoulder function score over time and groups. All statistical analyses were performed using SPSS version 25 software (IBM Corp., Armonk, NY, USA), and statistical significance was at *p* < 0.05.

## 3. Results

### 3.1. Participants’ Characteristics

Thirty-two patients (15 total mastectomy (TM), 17 partial mastectomy (PM)) diagnosed with breast cancer stage 0–3 participated in this study. The mean age of participants was 52.3 ± 7.6 years, and the mean body mass index was 24.6 ± 2.9 kg/m^2^. Of the participants, 46.9% underwent neoadjuvant chemotherapy (Table 1).

### 3.2. Change in Shoulder Range of Motion (from Pre-Surgery to 4-Week Post-Surgery)

The ROM of flexion and extension on the affected side was significantly reduced immediately after surgery. Although the ROM slowly improved, it remained considerably below baseline levels (flexion −36.1% to −18.2%, extension −27.2% to −16%). Notably, the ROM of shoulder abduction significantly decreased immediately after surgery (−46.7% at POD1) and barely recovered until the third outpatient visit (−45.7% at POD24) to the baseline level (Table 2). In the unaffected side, the only significant difference in ROM occurred in flexion and abduction on POD1, compared with the baseline ROM. After surgery, the ROM between the affected and unaffected arms exhibited a significant difference at all measurement points (Table 2).

When ROM was examined according to the surgical types, the greatest reduction in the affected side occurred in patients who underwent TM with axillary lymph node dissection (ALND) (flexion −48.8% to −23.5%, abduction −61.1% to −51.2%). The smallest reduction was observed in those who underwent PM with SLNB (PM&SLNB; flexion −19.9% to −8.1%, abduction −29.5% to −27.2%). However, no significant difference existed among the groups and time effects (Figure 1).

### 3.3. Change in Shoulder Strength (from Pre-Surgery to 4-Week Post-Surgery)

Shoulder strength of the affected side was significantly reduced immediately after surgery (reduction rate relative to baseline: flexion −50.6%, extension −44.8%, abduction −49.1%, horizontal adduction −41.3%, and horizontal abduction −36.8%) and slowly recovered in the 24 days after surgery. In contrast, shoulder strength of the unaffected side gradually decreased over time. Except for shoulder extension on POD10, significant differences between the affected and unaffected arms were observed at all measurement points after surgery (Table 3). A similar pattern of reduction was observed in the shoulder strength of the affected side regardless of the surgical methods (Figure 2).

### 3.4. Change in Shoulder Function (from Pre-Surgery to 4-Week Post-Surgery)

On the affected arm, both shoulder ROM and strength scores significantly decreased compared to their pre-surgery levels and did not fully recover until 4 weeks after surgery (Table 4). On the unaffected arm, the ROM score only decreased on POD1 compared to its pre-surgery level. Conversely, the shoulder strength of the unaffected arm significantly reduced throughout the study period compared to its pre-surgery level. When shoulder recovery patterns were analyzed by age, body mass index (BMI), and the number of lymph nodes removed, recovery patterns did not differ by BMI and number lymph node recovery: only age (under 55 age vs. over 55 age) showed a difference in the recovery of the ROM score of the affected side (*P* for time = 0.04; Appendix A).

The ROM score showed a different declining pattern on the affected arm according to the surgical method. In terms of strength, a similar pattern of reduction was observed regardless of the surgery method (Figure 3). 

## 4. Discussion

Previous studies have recommended early exercise intervention to alleviate shoulder discomfort and dysfunction related to breast cancer surgery. However, there is insufficient evidence of the extent of the recovery of shoulder ROM and strength during the early recovery phases, such as from immediately after surgery to until 4 weeks. Furthermore, shoulder recovery during the early recovery phase based on different breast cancer surgeries has not been studied. Therefore, we investigated shoulder ROM and strength after breast cancer surgery according to different surgical methods. We found a significant reduction in shoulder ROM immediately after surgery (36.7% reduction compared to baseline), which recovered to up to 73.5% of the pre-surgery level. A significant reduction in shoulder ROM after 4 weeks was observed in patients regardless of the surgical method; however, patients who underwent TM with ALND seemed to have the most decreased shoulder ROM. Shoulder ROM on the unaffected side was also reduced when measured a day after surgery; however, this phenomenon was not observed 4 weeks after surgery. Similarly, we further noticed that shoulder strength was significantly decreased after surgery (48.1% reduction compared to baseline), which recovered only up to 62.5% at 4 weeks after surgery. The degree of reduction in shoulder strength was similar regardless of the surgical method. Unlike shoulder ROM, a significant decrease in shoulder strength was also observed in the unaffected arm at 4 weeks after surgery (78.9% compared to baseline).

Within the first month after breast cancer surgery, patients experienced the greatest change in ROM. In this study, we found a 36.7% reduction in shoulder ROM at POD1, which recovered to up to 73.5% of the pre-surgery levels at 1 month after surgery. In contrast, the ROM of the unaffected arm was significantly reduced on POD1 and recovered fully by POD10. The significant reduction in shoulder ROM at 1 month after surgery observed in our study agrees with a previously reported study. Springer et al. reported that shoulder ROM recovered to up to about 93% of baseline. They further observed that participants’ shoulder ROM recovered to the pre-surgery level at 1 year after surgery [16]. Cinar et al. [17] reported a significant reduction in shoulder ROM on the fifth day after surgery (treatment group: flexion 74.7%, abduction 64.6% of baseline; home exercise program group: flexion 60.5%, abduction 53.5% of baseline), which did not fully recover until 1 month after surgery (treatment group: flexion 95.8%, abduction 93.4% of baseline; home exercise program group: flexion 76.5%, abduction 69.4% of baseline). In our study, the ROM of the unaffected arm showed a reduction on POD1, which had almost recovered to the baseline level on POD10.

We further studied whether the recovery of shoulder ROM differs according to the surgical method. As expected, patients who underwent PM with SLNB seemed to recover their shoulder ROM better than those who underwent other methods of surgery, such as TM with ALND. Nesvold et al. [18] reported that impaired shoulder flexion (by ≥25° of the unaffected arm) was observed in 24% of patients who underwent radical modified mastectomy and 7% of those who underwent breast-conserving surgery. Compared to breast-conserving surgery, mastectomy significantly increased the risk of impaired shoulder ROM in flexion and abduction by 3.3 (OR 3.3, 95% CI 1.2–9.14; *p* = 0.02) and 2.3 (OR 2.3, 95% CI 1.06–4.98; *p* = 0.04) times, respectively. Participants who underwent ALND showed a considerable loss of shoulder ROM compared to the SLNB group, even after several years post-surgery [9,19].

There are a few studies which have observed the change in shoulder strength after breast cancer surgery. Belmonte et al. [20] reported significant reduction in shoulder strength in both the affected and unaffected arm 5 years post-surgery. Klassen et al. [21] also observed 12–16% lower shoulder strength among breast cancer patients during cancer treatments (e.g., surgery, chemotherapy, radiation, etc.). However, both studies observed changes in shoulder strength among breast cancer patients from a few months to a few years after breast cancer surgery, with changes immediately after surgery being absent. In our study, a significant reduction in shoulder strength was found in both the affected and unaffected arms immediately after surgery, which did not recover up to four weeks. Interestingly, shoulder strength was significantly reduced in not only the affected arm but also the unaffected arm, although the pattern of decline in shoulder strength differed. The strength of the affected arm decreased immediately after surgery and recovered only up to 52.9% of the pre-surgery level, while the strength of the unaffected arm gradually declined over 4 weeks.

We noticed that shoulder strength significantly decreased after surgery regardless of the surgical method; these results differ from those of previous studies. Belmonte et al. [20] and Sagen et al. [9] reported that participants who underwent ALND showed a slower recovery of shoulder strength than those who underwent SLNB. The reason why we did not observe differences in our participants’ shoulder strength recovery after surgery is unclear. It may be due to the lack of upper body use after surgery or to the relatively short follow-up duration in our participants. We followed up our participants for up to 4 weeks only, whereas other studies followed up their participants’ shoulder strength recovery up to several years [9,19,22]. If we had performed a longer follow-up of shoulder strength in our participants, we may have observed different shoulder strength recoveries based on the surgical methods.

The reduction in shoulder strength regardless of surgical methods (whether they are more invasive, which may cause more damages in muscles and facia, or less invasive, which may cause only a small or no damage in the muscles and facia) is of interest. In general, upper limb problems, including reduction in shoulder strength after mastectomy or axillary surgery, can cause lymphedema, shoulder restriction, pain, numbness, and weakness [22,23,24]. In the process of healing, damaged tissue and fascia could become shorter than their lengths before surgery, leading to limitations in shoulder movement [24,25]. Additionally, surgery carries the risk of causing nerve injury (involving the intercostobrachial nerve or long thoracic nerve), which can lead to sensation changes or chronic pain in the surgical area [26].

As mentioned above, a gradual reduction in shoulder strength on the unaffected side is an interesting finding. Theoretically, there is no anatomical and physiological reason for this reduction in strength over time. However, breast cancer patients seemed to be discouraged to use their upper body, including both the affected and unaffected sides. Therefore, reduction in shoulder strength could be due to both lack of use and the results of surgery, such as anatomical damage of muscle, facia, and skin. About 70% of breast cancer patients showed avoidance of strenuous arm activity after surgery, and the primary reasons were pain, scar formation, swelling and considerable fear of lymphedema, misinterpretation of arm care advice, and low coping ability [27]. In addition, patients who avoided using their arms reported more arm and chest symptoms than those who did not avoid using their upper limbs [27]. Patients experiencing pain after surgery may tend to adopt protective postures such as dropped and rounded shoulder and arm, in addition to reduced use of the arm. This results in long-term changes to muscle length and activity [28]. The various physical and psychological effects of surgery can also impede the return to activities of daily living and increase long-term stress [20,29]. 

This study had some limitations. First, the sample size was small and, therefore, it is difficult to generalize our results. Second, more than 80% of participants were less than 60 years of age and, therefore, our results may not be generalizable to older breast cancer patients (>60 years). Third, when we measured ROM and strength, participants’ shoulder problems before surgery and physical activities after surgery were not controlled. Despite these limitations, we believe that the current study has important clinical implications for understanding the patterns of shoulder function, including shoulder ROM and strength, in the early recovery phase in post-surgery breast cancer patients.

The main finding of this study indicates that shoulder ROM and strength decreased in postoperative breast cancer patients. The reduction in shoulder ROM was observed after surgery, especially so on the affected arm rather than the unaffected arm. The shoulder strength was reduced in both the affected and unaffected sides after surgery. The recovery of ROM showed different patterns depending on the surgical method; however, the strength of the shoulder showed a similar pattern of decrease regardless of the surgical methods. The results of this study suggest the need for early rehabilitation after breast cancer surgery.

## Figures and Tables

**Figure 1 jcm-10-03416-f001:**
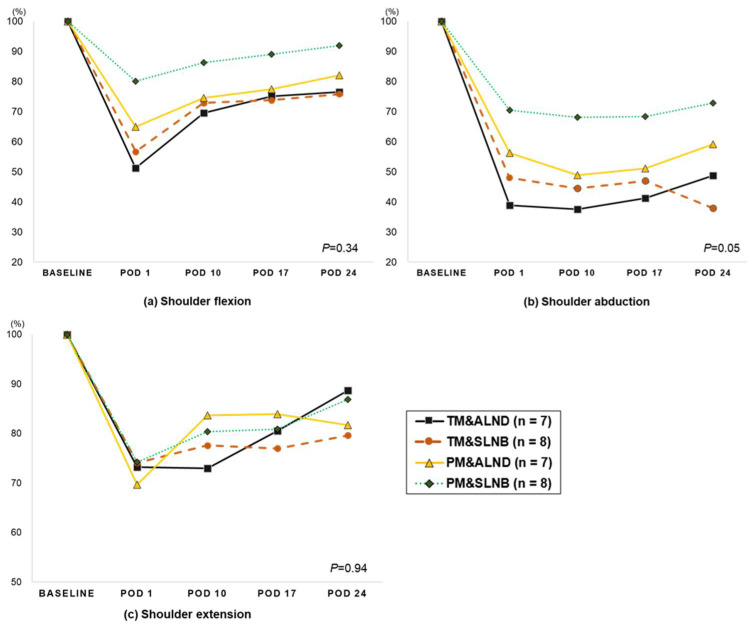
The changes in the shoulder range of motion after breast surgery according to the surgery type in the affected side (descriptive). *P*: the repeated measure of ANOVA analysis (group time interaction). Abbreviation: Total mastectomy with axillary lymph node dissection; TM&ALND, Total mastectomy with sentinel node biopsy; TM&SLNB, Partial mastectomy with axillary node dissection; PM&ALND, Partial mastectomy with sentinel node biopsy; PM&SLNB, Postoperative day: POD.

**Figure 2 jcm-10-03416-f002:**
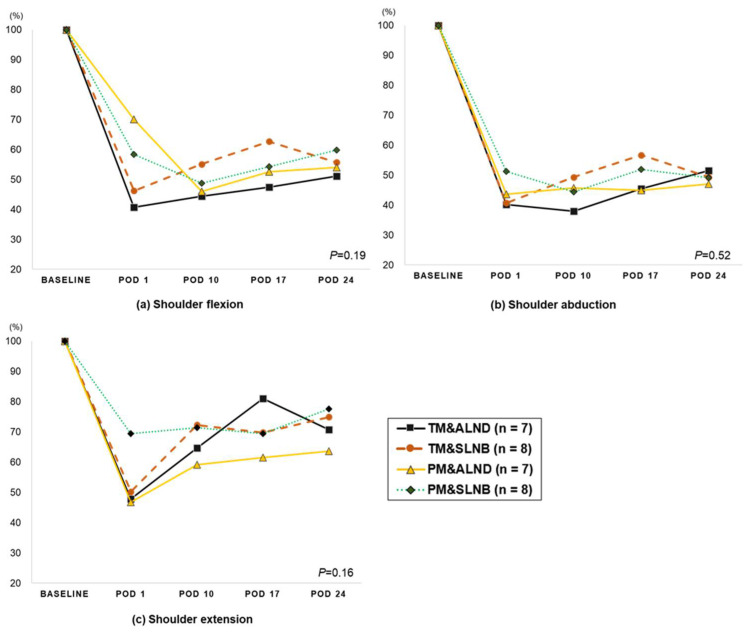
The changes in the shoulder strength after breast surgery according to the surgery type in the affected side (descriptive). *P*: the repeated measure of ANOVA analysis (group*time interaction). Abbreviation: Total mastectomy with axillary lymph node dissection; TM with ALND, Total mastectomy with sentinel node biopsy; TM c SLNB, Partial mastectomy with axillary node dissection; PM with ALND, Partial mastectomy with sentinel node biopsy; PM with SLNB, Postoperative day: POD.

**Figure 3 jcm-10-03416-f003:**
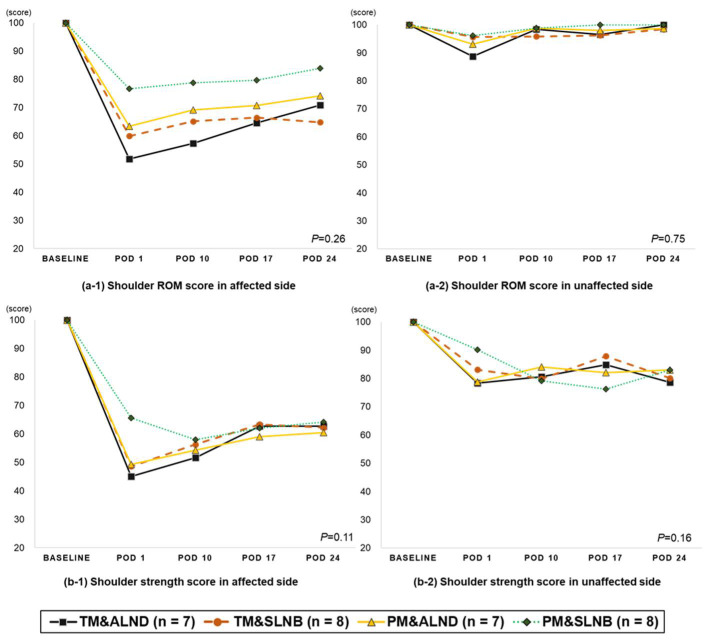
The changes in the shoulder function score in breast cancer patients after surgery according to the surgery type (descriptive). *P*: the repeated measure of ANOVA analysis (group*time interaction). Abbreviation: Total mastectomy with axillary lymph node dissection; TM with ALND, Total mastectomy with sentinel node biopsy; TM c SLNB, Partial mastectomy with axillary node dissection; PM with ALND, Partial mastectomy with sentinel node biopsy; PM with SLNB, Postoperative day: POD.

**Table 1 jcm-10-03416-t001:** Participants’ characteristics.

Variable (*N* = 32)	TM&ALND (*N* = 7)	TM&SLNB (*N* = 8)	PM&ALND (*N* = 8)	PM&SLNB (*N* = 9)	*P*
Age (years)	50.0 ± 11.1	53.6 ± 6.7	52.6 ± 7.9	52.8 ± 5.3	0.8
Weight (kg)	61.7 ± 8.3	66.5 ± 9.7	63.0 ± 7.9	58.4 ± 6.3	0.3
BMI (kg/m^2^)	23.3 ± 2.4	25.9 ± 3.7	25.5 ± 2.9	23.7 ± 1.9	0.2
Muscle mass (kg)	23.0 ± 2.3	23.6 ± 2.7	22.4 ± 2.3	20.8 ± 2.1	0.1
Fat (%)	30.1 ± 4.6	33.5 ± 5.5	33.0 ± 6.0	33.6 ± 6.6	0.6
Stage (n, %)					
0	0	3 (37.5%)	2 (25%)	4 (44.4%)	0.6
1	1 (14.3%)	3 (37.5%)	2 (25%)	5 (55.6%)	
2	3 (42.9%)	2 (25%)	3 (37.5%)	0	
3	3 (42.9%)	0	1 (12.5%)	0	
Surgery site (n, %)					
Right	5 (71.4%)	4 (50%)	4 (50%)	6 (66.7%)	0.8
Dominant arm	5 (71.4%)	5 (62.5%)	4 (50%)	6 (66.7%)	0.8
Dissected LNs	18 (9–25)	7 (3–13)	17.5 (10–26)	9 (5–12)	<0.001
Surgery duration (min)					
	108.9 ± 21.4	95 ± 16.1	122.3 ± 59.4	93.4 ± 40.0	0.5
Drainage removal day (n, %)					
1st outpatient visit (POD 10)	1 (14.3%)	1 (12.5%)	0	3 (33.3%)	0.3
2nd outpatient visit (POD 17)	6 (85.7%)	4 (50%)	4 (50%)	4 (44.4%)	
3rd outpatient visit (POD 24)	0	3 (37.5%)	3 (37.5%)	1 (11.1%)	
4th outpatient visit	0	0	1 (12.5%)	1 (11.1%)	
Neoadjuvant chemotherapy					
Yes	6 (85.7%)	1 (12.5%)	7 (87.5%)	1 (11.1%)	<0.001

Values are presented as Mean ± SD or n (%), Dissected LNs are showed median (minimum to maximum range). Abbreviation: Total mastectomy with axillary lymph node dissection; TM with ALND, Total mastectomy with sentinel node biopsy; TM with SLNB, Partial mastectomy with axillary node dissection; PM with ALND, Partial mastectomy with sentinel node biopsy; PM with SLNB, Lymph Nodes; LNs, Postoperative Date; POD.

**Table 2 jcm-10-03416-t002:** Change of shoulder range of motion (pre-surgery to 4 weeks post-surgery).

	Baseline	POD 1	POD 10	POD 17	POD 24	*P ^#^*	*P ^baseline^* ^vs. *POD1*^	*P ^POD1^* ^vs. *POD10*^	*P ^POD10^* ^vs. *POD17*^	*P ^POD17^* ^vs. *POD24*^
Shoulder range of motion										
Flexion										
Affected side (n = 29)	174.0 ± 1.3	111.1 ± 8.6 **+	132.5 ± 4.7 **+	137.5 ± 4.2 **+	142.3 ± 4.6 **+	<0.001	<0.001	0.005	0.071	0.01
Unaffected side (n = 31)	176.4 ± 0.8	161.7 ± 3.6 **	174.4 ± 0.7	174.7 ± 0.9	174.9 ± 0.7	0.02	<0.001	0.001	0.765	0.625
Abduction										
Affected side (n = 30)	169.1 ± 2.8	90.2 ± 7.1 **+	84.1 ± 4.5 **+	87.9 ± 5.7 **+	91.8 ± 6.1 **+	<0.001	<0.001	0.386	0.437	0.164
Unaffected side (n = 31)	170.2 ± 1.9	153.7 ± 4.4 *	166.0 ± 2.4	169.0 ± 2.1	171.2 ± 2.1	0.004	0.001	0.0.21	0.197	0.05
Extension										
Affected side (n = 30)	47.4 ± 1.1	34.5 ± 2.1 **+	37.3 ± 1.6 **+	38.1 ± 1.8 **+	39.8 ± 1.6 **+	<0.001	<0.001	0.200	0.642	0.101
Unaffected side (n = 31)	46.0 ± 1.2	44.7 ± 1.8	44.1 ± 1.5	43.8 ± 1.6	45.8 ± 1.2	0.36	0.568	0.788	0.859	0.133

Values are presented as mean ± standard error (SE). Postoperative Date; POD, *p*
^#^ represent overall differences according to time as determined using the repeated measures of ANOVA, * *p* < 0.01 vs. Baseline, ** *p* < 0.001 vs. Baseline, **^+^**
*p* < 0.01 between the affected and unaffected side (POD1 to POD 24 adjusted for Baseline value).

**Table 3 jcm-10-03416-t003:** Change of shoulder strength (pre-surgery to 4 weeks post-surgery).

	Baseline	POD 1	POD 10	POD 17	POD 24	*P ^#^*	*P ^baseline^* ^vs. *POD1*^	*P ^POD1^* ^vs. *POD10*^	*P ^POD10^* ^vs. *POD17*^	*P ^POD17^* ^vs. *POD24*^
Shoulder strength										
Flexion										
Affected side (n = 31)	16.0 ± 0.9	7.9 ± 0.8 **+	7.8 ± 0.7 **+	8.8 ± 0.7 **+	8.9 ± 0.6 **+	<0.001	<0.001	0.890	0.017	0.850
Unaffected side (n = 31)	15.5 ± 0.8	12.7 ± 0.9 **	11.8 ± 0.8 **	11.7 ± 0.6 **	11.0 ± 0.5 **	<0.001	<0.001	0.173	0.866	0.071
Abduction										
Affected side (n = 31)	15.5 ± 1.0	7.0 ± 0.6 **+	7.0 ± 0.6 **+	7.8 ± 0.7 **+	7.6 ± 0.5 **+	<0.001	<0.001	0.992	0.009	0.422
Unaffected side (n = 31)	15.0 ± 0.9	12.1 ± 0.9 **	11.2 ± 0.7 **	11.1 ± 0.6 **	11.0 ± 0.6 **	<0.001	<0.001	0.181	0.823	0.607
Extension										
Affected side (n = 31)	22.3 ± 1.0	12.3 ± 1.0 **+	15.1 ± 1.0 **	15.5 ± 0.8 **+	16.1 ± 0.7 **+	<0.001	<0.001	0.001	0.426	0.521
Unaffected side (n = 31)	22.7 ± 1.2	17.6 ± 1.1 **	17.4 ± 1.0 **	19.0 ± 0.9 **	18.0 ± 0.8 **	<0.001	<0.001	0.783	0.047	0.147
Horizontal adduction										
Affected side (n = 31)	19.6 ± 1.0	11.5 ± 1.1 **+	11.3 ± 1.0 **+	12.3 ± 0.9 **+	12.9 ± 1.0 **+	<0.001	<0.001	0.754	0.073	0.355
Unaffected side (n = 31)	19.5 ± 1.1	16.4 ± 1.0 **	16.7 ± 1.2 *	16.7 ± 0.9 *	15.3 ± 0.8 **	<0.001	<0.001	0.649	0.979	0.006
Horizontal abduction										
Affected side (n = 30)	19.3 ± 1.2	12.2 ± 1.1 **+	11.5 ± 1.0 **+	12.7 ± 0.8 **+	12.0 ± 0.8 **+	<0.001	<0.001	0.312	0.066	0.263
Unaffected side (n = 31)	19.2 ± 1.0	17.4 ± 1.1	15.9 ± 1.2 *	15.6 ± 0.8 **	14.9 ± 0.6 **	<0.001	0.020	0.025	0.771	0.095

Values are presented as mean ± standard error (SE). Postoperative Date; POD, *P*
^#^ represent overall differences according to time as determined using the repeated measures of ANOVA, * *p* < 0.01 vs. Baseline, ** *p* < 0.001 vs. Baseline, **^+^**
*p* <0.01 between the affected and unaffected side (POD1 to POD 24 adjusted for Baseline value).

**Table 4 jcm-10-03416-t004:** Change of shoulder function score (pre-surgery to 4 weeks post-surgery).

	Baseline	POD 1	POD 10	POD 17	POD 24	*P ^#^*	*P ^baseline^* ^vs. *POD1*^	*P ^POD1^* ^vs. *POD10*^	*P ^POD10^* ^vs. *POD17*^	*P ^POD17^* ^vs. *POD24*^
Shoulder function score										
ROM score										
Affected side (n = 30)	100.0 ± 0	63.3 ± 4.0 **+	67.8 ± 2.4 **+	70.5 ± 2.4 **+	73.5 ± 2.6 **+	<0.001	<0.001	0.208	0.146	0.011
Unaffected side (n = 31)	100.0 ± 0	93.6 ± 2.4 *	97.9 ± 1.4	98.5 ± 1.5	100.4 ± 1.3	0.018	0.012	0.072	0.705	0.085
Strength score										
Affected side (n = 31)	100.0 ± 0	52.9 ± 2.9 **+	55.3 ± 2.2 **+	61.8 ± 2.1 **+	62.5 ± 2.3 **+	<0.001	<0.001	0.369	0.001	0.964
Unaffected side (n = 31)	100.0 ± 0	83.1 ± 2.3 **	80.8 ± 2.8 **	82.5 ± 2.4 **	78.9 ± 2.9 **	<0.001	<0.001	0.335	0.461	0.035

Values are presented as mean ± standard error (SE), abbreviation: Postoperative Date; POD. *P*
^#^ represent overall differences according to time as determined using the repeated measures of ANOVA, * *p* < 0.01 vs. Baseline, ** *p* < 0.001 vs. Baseline, + *p* < 0.01 between the affected and unaffected side (POD1 to POD 24 adjusted for Baseline value).

## Data Availability

The data in the current study are available on request from the corresponding author. The data are not publicly available due to the ethical considerations.

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
