# Peer review of "Change in Shoulder Function in the Early Recovery Phase after Breast Cancer Surgery: A Prospective Observational Study"

_jcm, 2021, doi:10.3390/jcm10153416_

Round 1

Reviewer 1 Report

The following aspects should be taken into consideration:

  • line 78 „participants were” should be rephrased;
  • 79: the period should be states as 1st February to 30 th November or the exact period of the study
  • 79-80: all the eligible criteria should be present or only one?
  • 82: all the exclusion criteria should be present or only one ?
  • 97: three shoulder moves or shoulder movement - would be more accurate 
  • 117: please specify the value of α
  • 124: I think it is more accurate to say 95% confidence interval or level of confidence
  • is there any correlation between the loss of motion in patient who underwent chemotherapy? especially in the healthy side?

The following should be revised : the following citation - to be adapted to the content of the present article;

 Cho, Y.; Do, J.; Jung, S.; Kwon, O.; Jeon, J. Y., Effects of a physical therapy program combined with manual lymphatic 344 drainage on shoulder function, quality of life, lymphedema incidence, and pain in breast cancer patients with axillary web 345 syndrome following axillary dissection. Support Care Cancer 2016, 24 (5), 2047-2057. 

-  the information in paragraph 59-60 it is not found in this article - study period - 4 weeks after intervention; it is however found in citation 9.

- if they saw that previous studies  recommend physical therapy after breast surgery for the recovery of the upper limb motion,  why not recommend it for the patients included in this study and after that the results would have been analysed ? in fact they concluded that 306-307: „The results of this study suggest the need for early rehabilitation after breast cancer surgery. ”

- the same conclusion was present in article above : 

„PT improves shoulder function, pain, and QOL in breast cancer patients with AWS and combined with MLD decreases arm lymphedema.” (https://pubmed.ncbi.nlm.nih.gov/26542271/ )

- they should mention why they did not include in the study design postoperative physical therapy

Overall, this article is well structured but it should be modified in order to comply with the standards of this outstanding journal.

Author Response

Reviewer #1

Thank you for your thoughtful comment and we appreciate the careful review of our manuscript.

We have addressed your comments below.

Comment #1

             line 78 participants were” should be rephrased;

Response #1

             According to the comment, we revised the manuscript as follows:

             “32 breast cancer patients were recruited”

Comment #2

             79: the period should be states as 1st February to 30 th November or the exact period of the study

Response #2

             As suggested, we have revised our manuscript as follows:

             “from 14th February to 2nd November 2019”

Comment #3

             79-80: all the eligible criteria should be present or only one?

Response #3

             There are three eligibility criteria and those who met all eligibility criteria we recruited for the current study. We have presented all three criteria in the manuscript as follows:

“Eligibility criteria included the following: 1) age between 19 and 70 years; 2) histologically confirmed stage <â…£ breast cancer; and 3) ability to understand and provide written informed consent in Korean.”

Comment #4

             82: all the exclusion criteria should be present or only one?

Response #4

             We excluded patients who one any of the three exclusion criteria. To improve the clarity of our manuscript, we have revised our manuscript as follows: 

We excluded 1) patients who were scheduled bilateral breast surgery, 2) breast reconstruction surgery or 3) existing evidence of recurrent or metastatic diseases.”

Comment #5

             97: three shoulder moves or shoulder movement - would be more accurate 

Response #5

             Thank you for your kind comments. According to your advice, we change the word as follows.

                 “Three shoulder movements”

Comment #6

             117: please specify the value of α

Response #6

             As suggested, we have revised and added our manuscript as below:

             “Statistical significance was at p<0.05.”

Comment #7

             124: I think it is more accurate to say 95% confidence interval or level of confidence

Response #7

             Thank you for your thoughtful advice. The sentence was written to represent α-value.

             It has been revised as follows to convey clear meaning:

             “Statistical significance was at p<0.05.”

Comment #8

             is there any correlation between the loss of motion in patient who underwent chemotherapy? especially in the healthy side?

Response #8

Thank you for your insightful comment.

             In this study, patients who received neoadjuvant tended to have lower preoperative strength than other breast cancer patients. However, the recovery of shoulder ROM and strength after surgery did not show significant differences. It is difficult to generalize due to the small sample size of this study. It is necessary to recruit more subjects in a future study to examine the difference in the recovery of shoulder function depending on whether neoadjuvant chemotherapy.

Comment #9

             The following should be revised: the following citation - to be adapted to the content of the present article;

             Cho, Y.; Do, J.; Jung, S.; Kwon, O.; Jeon, J. Y., Effects of a physical therapy program combined with manual lymphatic 344 drainage on shoulder function, quality of life, lymphedema incidence, and pain in breast cancer patients with axillary web 345 syndrome following axillary dissection. Support Care Cancer 2016, 24 (5), 2047-2057. 

             -  the information in paragraph 59-60 it is not found in this article - study period - 4 weeks after intervention; it is however found in citation 9.

Response #9

             We thank to the reviewer for this very important comment. As pointed out by the reviewer, the part where the reference was incorrectly marked was reconfirmed and corrected. We appreciated your careful review.

Comment #10

             - if they saw that previous studies recommend physical therapy after breast surgery for the recovery of the upper limb motion, why not recommend it for the patients included in this study and after that the results would have been analysed ? in fact they concluded that 306-307: „The results of this study suggest the need for early rehabilitation after breast cancer surgery. ”

             - the same conclusion was present in article above : 

             „PT improves shoulder function, pain, and QOL in breast cancer patients with AWS and combined with MLD decreases arm lymphedema.” (https://pubmed.ncbi.nlm.nih.gov/26542271/ )

             - they should mention why they did not include in the study design postoperative physical therapy

             Overall, this article is well structured but it should be modified in order to comply with the standards of this outstanding journal.

Response #10

             We thank to the reviewer for this very important comment.

             Unfortunately, in Korea, it is rare for breast cancer patients to receive physical therapy immediately after surgery due to the lack of evidence for early rehabilitation (e.g., increased drainage volume, lymphedema risk issue, surgical infection risk). Patients are discharged 1-2 days after surgery and then receive outpatient treatment once a week for one month. If there is a shoulder problem in the subsequent process, it is common to receive various rehabilitation, including manual lymphatic drainage. None of the subjects included in this study did receive early rehabilitation immediately after breast cancer surgery due to the study.

             Although studies on early rehabilitation immediately after breast cancer surgery have been continuously reported, studies on the recovery trend of shoulder function according to surgical methods during the early recovery phase are very scarce. Therefore, this study provided the level of recovery of upper body function in breast cancer patients according to the surgery method in early recovery period. The results of this study can be used as reference data for early rehabilitation after breast cancer surgery.

Reviewer 2 Report

My comment refers to the edition of the article, to improve the interpretation of the figures.

I suggest that each line be of one color, in order to better distinguish the results of each series of patients.

Author Response

 Thank you for your thoughtful advice. As suggested, we have revised figure.
